# Identification of the Key miRNAs and Genes Associated with the Regulation of Non-Small Cell Lung Cancer: A Network-Based Approach

**DOI:** 10.3390/genes13071174

**Published:** 2022-06-29

**Authors:** Zoya Shafat, Mohd Murshad Ahmed, Fahad N. Almajhdi, Tajamul Hussain, Shama Parveen, Anwar Ahmed

**Affiliations:** 1Centre for Interdisciplinary Research in Basic Sciences, Jamia Millia Islamia, New Delhi 110025, India; zoya179695@st.jmi.ac.in (Z.S.); mahmed4@jmi.ac.in (M.M.A.); sparveen2@jmi.ac.in (S.P.); 2Centre of Excellence in Biotechnology Research, College of Science, King Saud University, Riyadh 11451, Saudi Arabia; majhdi@ksu.edu.sa (F.N.A.); thussain@ksu.edu.sa (T.H.); 3Department of Botany & Microbiology, College of Science, King Saud University, Riyadh 11451, Saudi Arabia

**Keywords:** non-small cell lung cancer, differentially expressed miRNAs, miRNA–mRNA network, module detection, gene term enrichment analysis, survival analysis, transcription factor

## Abstract

Lung cancer is the major cause of cancer-associated deaths across the world in both men and women. Lung cancer consists of two major clinicopathological categories, i.e., small cell lung cancer (SCLC) and non-small cell lung cancer (NSCLC). Lack of diagnosis of NSCLC at an early stage in addition to poor prognosis results in ineffective treatment, thus, biomarkers for appropriate diagnosis and exact prognosis of NSCLC need urgent attention. The proposed study aimed to reveal essential microRNAs (miRNAs) involved in the carcinogenesis of NSCLC that probably could act as potential biomarkers. The NSCLC-associated expression datasets revealed 12 differentially expressed miRNAs (DEMs). MiRNA-mRNA network identified key miRNAs and their associated genes, for which functional enrichment analysis was applied. Further, survival and validation analysis for key genes was performed and consequently transcription factors (TFs) were predicted. We obtained twelve miRNAs as common DEMs after assessment of all datasets. Further, four key miRNAs and nine key genes were extracted from significant modules based on the centrality approach. The key genes and miRNAs reported in our study might provide some information for potential biomarkers profitable to increased prognosis and diagnosis of lung cancer.

## 1. Introduction

The highest number of deaths related to cancer is still associated with lung cancer across the globe, which results in around (or more than) 1.5 million deaths annually [1]. On the basis of treatment purpose, lung cancer has been categorized into two major subgroups, i.e., small cell cancer and non-small cell cancer. The second subgroup non-small cell lung cancer (NSCLC) constitutes majorly three histological subtypes, which are adenocarcinoma (40–50%), squamous cell carcinoma (around 30%), and large cell carcinoma (around 15%), which account for approximately 80% cases of lung cancer [1,2]. COPD (chronic obstructive pulmonary disease), and other type of lung disease like pulmonary fibrosis or pleural effusion could also be caused by NSCLC [3]. NSCLC occurrence is associated with signaling pathways (mTOR [4] and tyrosine kinase [5]) and oxidative stress [6] and is also related to the changes in the cell cycle. The existing treatment mainly involves platinum bimodal therapy (cytotoxic therapy) [7], however, in some patients, resistances to this therapy have been reported recently. Regardless of recent improvements in treatment, NSCLC is generally diagnosed at a highly developed (late) stage which results in low survival rates, due to poor prognosis [8]. In light of this, the identification of appropriate treatment strategies or novel diagnostic biomarkers is essential in controlling lung cancer.

The microRNAs (miRNAs) are small (~22 nucleotides) noncoding RNAs which regulate more than half of the genes in human cells [9]. An miRNA is linked with diverse biological activities which include cell differentiation, cell proliferation, disease initiation, cell migration, disease progression, and finally apoptosis [9]. The miRNA modulates the activity of genes at the level of posttranscription, by inhibiting their messenger RNA (mRNA) targets translation [10]. It was revealed that the expression for miRNAs is upregulated (frequently) for oncogenic miRNAs, while the downregulated expression of miRNA has been documented for tumor suppressor miRNA [10]. Studies have suggested that miRNAs perform a vital role in NSCLC development by acting as potential biomarkers in its diagnosis and prognosis [8,11,12,13]. Typically, the incidence and advancement of NSCLC are due to multistep carcinogenesis which involves various signal transduction pathways and change of gene expression levels [14,15]. The mechanisms leading to the promotion of carcinogenesis in NSCLC need to be exploited. Recently, several reports have documented the role of various miRNAs expression in different cancers (including NSCLC), probably, suggesting their crucial roles in carcinogenesis [14,16,17]. In particular, some miRNAs (miRNA-224, miRNA-30d-5p) have also been demonstrated to play important role in NSCLCs as either promoters or suppressor (cancer promoters or as cancer suppressors) [18,19].

Though these aforementioned studies have documented the role of miRNAs in lung cancer or specifically NSCLC, these reports comprised different datasets [8,9,10,11,12,13,14,15,16,17,18,19]. Therefore, in this regard, our present study performed an integrated analysis on some of the other unexplored gene expression profiles of NSCLC. Thus, our identified key miRNAs and related genes show a discrepancy with the previous study results due to heterogeneity in NSCLC cases and control subjects. In this study, we used network analyses to show the correlation between the identified key miRNAs/genes and NSCLC. This kind of study is envisaged to provide useful information in exploring candidate miRNA biomarkers in human NSCLC.

The present study reports the analysis of the identified signature miRNAs between three distinguishing NSCLC series (GSE25508, GSE19945, and GSE53882) using a bioinformatics approach. Our study revealed several promising key miRNAs/genes that have been constantly reported in lung cancer-associated profiling studies. Their key candidates might provide some information about miRNA’s role in tumorigenesis and its related mechanisms. The GEO (Gene Expression Omnibus) datasets were investigated to obtain miRNAs that were differentially expressed between non-small cell lung cancer tissues and normal tissue samples. A comparative analysis was undertaken to select the differentially expressed miRNAs (DEMs) among these retrieved datasets. To locate the DEMs associated with target genes the RNA interactome encyclopedia was used. Further, network analysis was applied to identify DEMs, which were then combined with mRNA to form the mRNA–miRNA network, to elucidate key miRNAs as well as their genes. Moreover, an enrichment analysis was performed for these key elements (key miRNAs and key genes) was explored to reveal their potential molecular mechanisms in NSCLC. Subsequently, the expression and validation analysis was applied to key genes. The obtained key genes regulated by miRNAs may provide some clue about the potential biomarkers profitable to increased prognosis and diagnosis of lung cancer. Therefore, these genes/miRNAs might be explored in therapeutic interventions of NSCLC after appropriate validation.

## 2. Materials and Methods

The graphical illustration of the network-based integrative method used in the current study is represented in Figure 1.

### 2.1. Search Strategy and Inclusion Criteria of Studies

We searched the GEO database (https://www.ncbi.nlm.nih.gov/geo/, accessed on 1 June 2021) for publicly available studies using the following keywords: “microRNA expression or miRNA expression”, “lung cancer or NSCLC”, “prognosis”, “non-small cell lung”, “adenocarcinoma”, “squamous cell carcinoma”, “large cell carcinoma” and “*Homo sapiens*” (organism). After a systematic and extensive review, we retrieved three GSE series. The criteria for inclusion of miRNA series included: (1) samples included normal tissue samples as well as diagnosed ones (NSCLC tissue samples), (2) miRNA expression profilings, (3) the minimum limit of the sample count in each group was 3, and (4) adequate information was collected to perform this research. These obtained miRNA expression profiles (GSE25508, GSE19945, and GSE53882) were used for the present analysis.

### 2.2. Acquisition of MiRNA Expression Data

The miRNA expression profiles of GSE25508, GSE19945, and GSE53882 were retrieved from the GEO database of the National Centre for Biotechnology Information (NCBI) [20]. These aforementioned expression series were generated from GPL7731 (Agilent 019,118 Human miRNA Microarray 2.0 G4470B), GPL9948 (Agilent Human 0.6K miRNA Microarray G4471), and GPL18130 (State Key Laboratory Human microRNA array 1888) platforms respectively. The expression dataset GSE25508 consisted of 34 lung cancer and 26 normal lung tissue samples. GSE19945 expression dataset consisted of 55 lung cancer and 8 noncancer lung tissue samples. The final expression dataset GSE53882 consisted of 151 patients with NSCLC and 397 corresponding adjacent noncancerous tissues.

### 2.3. Data Preprocessing and Screening of Differentially Expressed MiRNAs (DEMs)

The GSE series were normalized and preprocessed through GEO2R, web-based analytical tool (http://www.ncbi.nlm.nih.gov/geo/geo2r/ accessed on 1 June 2021). It constitutes Linear Models for Microarray Data (Limma) R package and GEO query. The preprocessing of datasets was undertaken to utilize default parameters. Benjamini-Hochberg correction method was used to correct the significant *p*-values obtained by the original hypothesis test. The differentially expressed microRNAs were extracted by applying the inclusion criteria: adjusted *p*-value (*p* < 0.05) and a |log_2_ (fold-change) > 1|. Overlapped DEMs among three miRNA series were obtained by Venny 2.1.0 (http://bioinfogp.cnb.csic.es/tools/venny/, accessed on 1 June 2021). It is an online tool which finds the intersection(s) of listed elements.

### 2.4. Identification of the DEM Target Genes

For the prediction of the target genes (associated with DEMs), four different databases were used. (1) TargetScan (http://www.targetscan.org/vert_72/, accessed on 1 July 2021), an algorithm, thatpredicts miRNA targets by comparing multiple genomes [21]. (2) miRmap (https://mirmap.ezlab.org/, accessed on 1 July 2021), a freely available (open source) Python library, includes web facility to predict miRNAtargets [22]. (3) miRWalk (http://mirwalk.umm.uni-heidelberg.de/, accessed on 1 July 2021) (version 3.0), a computational-based approach, predicts target sites encoded by Perl programming language [23]. (4) mirDIP (http://ophid.utoronto.ca/mirDIP/, accessed on 1 July 2021), a database, provides dependable, user-friendly, and inclusive resources to identify miRNAtargets [24]. The genes that were found to be overlapping in all the four databases were predicted as the target gene. Venny 2.1.0 (https://bioinfogp.cnb.csic.es/tools/venny/, accessed on 1 July 2021), online visualization software, was applied for the generation of the Venn diagram.

### 2.5. DEM–mRNA Network Construction

The miRNA–mRNA network was built by utilizing overlapped genes (target genes vs. CTD (Comparative Toxicogenomics Database) NSCLC genes) (Appendix A) and DEMs in Cytoscape (Version 3.7.1) software, manually using SIF files. The Cytoscape plugin cytoHubba (version 0.1) was exploited to identify significant modules, subnetworks, and top-ranked genes/nodes in a given network, by employing various topological algorithms. The overlapped nodes in four clustering methods were extracted. Finally, the obtained extracted nodes possessed hub genes and miRNAs.

### 2.6. Network Analysis

In the miRNA-mRNA network, each node represented the gene/miRNA and edges represented the connection between nodes. The following topological properties in the constructed miRNA-mRNA network were analyzed to find out the important behaviors of the network and hub nodes [25].

Degree distribution: In a particular network, the degree (*k*) of node reflects the total number of edges (connections) by which it is connected with other nodes [26]. The degree *k* of a node is a local measure of centrality of that node [26]. The degree distribution *P*(*k*) of a node *n* is given by the expression:$$P\left(k\right)=\frac{n_{k}}{N}$$
where, *n_k_* is the number of nodes having degree *k* and *N* is the total number of nodes in the network. *P(k)* indicates the importance of hubs or modules in the network.

Betweenness centrality: In a particular network, a node’s betweenness centrality reflects the importance of flow of information from one node to another based on the shortest path [27]. The betweenness centrality *C_B_(n)* of a node *n* is given by the expression:$$C_{B}\left(n\right)=\sum_{s\neq n\neq t}\frac{d_{st}\left(n\right)}{d_{sj}}$$
where, *s* and *t* are nodes in the network other than *n*, *d_st_* is the total number of shortest paths from *s* to *t*, and *d_st_* (*n*) is the number of those shortest paths from *s* to *t* on which *n* lies [26,28,29].

Closeness centrality: In a particular network, closeness centrality reflects how the information is rapidly passing from one node to another [30].
$$C_{C}\left(n\right)=\frac{N-1}{\sum_{J}d_{ij}}$$
where, *d_ij_* is the length of the shortest path between two nodes *i* and *j*, and *N* is the total number of nodes in the network which are connected to the node *n*.

Stress: In a particular network, stress reflects the addition of all nearest path of all node pairs [31]. In order to compute the stress of a node *v*, first calculate all shortest pathways in a graph G, then, count the number of shortest paths passing through *v*. A stressed node is the one that has large number of shortest paths passing through it. Notably, and may be more critically, a high stress number does not necessarily imply that node v is critical for maintaining the link between nodes whose pathways cross through it.

### 2.7. Gene Term Enrichment and Pathway Analysis

The pathway enrichment analysis of hub miRNAs was implemented using MIENTURNET (MIcroRNAENrichmentTURnedNETwork) web-tool (Mienturnet (uniroma1.it, accessed on 1 August 2021)) [32] that offers enriched KEGG (Kyoto Encyclopedia of Genes and Genomes) (https://www.genome.jp/kegg/, accessed on 1 August 2021) pathway visualization. Further, the identified key genes associated with miRNAs were accessed for their biological implications using gene ontology (GO) analysis. The GO was performed in these mentioned categories, the first one BP as biological process, second one CC as cellular component and the third one is MF as molecular function, using the GOnet server (a tool for interactive Gene Ontology analysis) (http://tools.dice-database.org/GOnet/, accessed on 1 August 2021).

### 2.8. Survival Analysis and Prediction of Transcription Factors (TFs)

GEPIA (Gene expression profiling interactive analysis) allows a user to interact with cancer and normal gene expression profiles (http://gepia.cancer-pku.cn/, accessed on 1 September 2021). The GEPIA data tool was exploited to examine the relation between expression of selected key genes and NSCLC prognosis. The survival analysis was undertaken by constructing the overall survival (OS) curve of key genes. The patients on the basis of gene expression (median) values were categorized into two classes. The OS of the key genes was evaluated by means of the Kaplan–Meier approach (using log-rank test) that provided survival plots. Furthermore, the key genes were validated using box plots and pathological stages were analyzed. The TRRUST (Transcriptional Regulatory Relationships Unraveled database was used to predict the TFs (http://www.grnpedia.org/trrust/, accessed on 1 September 2021).

## 3. Results

### 3.1. Selection of DEGs

The GSE series of NSCLC was denoted by X.

There are 3 GSE series of X (X1, X2 and X3)
Xu = (X_1_u_1_, X_2_u_2_, X_3_u_3_)(1)
where, u stands for upregulation.
Xd = (X_1_d_1_, X_2_d_2_, X_3_d_3_)(2)
where, d stands for downregulation.

To find DEMs, we compared equations, i.e., (1) with (2) as follows:∑X_u_ = X_1_u_1_∪X_2_u_2_∪X_3_u_3_  (Upregulated genes in three GSE series)(3)
∑X_d_ = X_1_d_1_∪X_2_d_2_∪X_3_d_3_  (Downregulated genes in three GSE series)(4)

To find combined DEMs, we merged Equations (3) with (4) as follows:∑Xud = ∑Xu∪∑Xd (Merge genes)(5)

Genes those showed values of *p* ≤ 0.05 along with log fold change of |0.5–2.0| were chosen as statistically significant (differentially expressed genes).

### 3.2. Selection of MiRNA from Datasets

According to search criteria, 3 NSCLC miRNA expression dataseries from published literature were retrieved from public databases. The description of datasets is provided in Table 1.

The miRNA expression profiles in these datasets were Venny tool compared. DEMs between cancer and normal tissue samples were identified in each GSE dataset. GSE25508 contained 39 upregulated and 52 downregulated miRNAs, GSE19945 consisted of 14 upregulated and 31 downregulated miRNAs and GSE53882 comprised 29 upregulated and 7 downregulated miRNAs. GSE25508 dataset had the largest number of upregulated miRNAs while GSE19945 dataset possessed the least number. Similarly, GSE25508 had the maximum downregulated miRNAs while GSE53882 had the lowest number of downregulated miRNAs. Therefore, the number of DEMs varied across the three studies. For identification of aberrant miRNAs associated with NSCLC, three aformentioned GEO datasets were utilized. The dataset contains 18,232 miRNAs in total, of which 172 DEMs were identified on the basis of fold change (>1.5) and *p*-value (<0.05). From these datasets, the 12 overlapped DEMs were identified of which 5 were upregulated and 7 were downregulated (Note: All miRNAs are mature miRNAs). The expression of these selected top ranked 12 DEMs is mentioned in Table 2.

### 3.3. Prediction of Target Genes for DEMs

MiRNA exerts its regulatory function through post-transcriptional silencing by binding to its complementary site on the target genes. The role of the obtained top-ranked 12 DEMs in NSCLC-associated pathogenesis was comprehended by identifying their target genes (OG vs. CTD) (Figure 2) (Appendix A). We identified the target genes for the top 12 DEMs using a combination of four databases, i.e., mirMap, TargetScan, miRWalk and mirDIP. Each of these databases showed different target genes for each of the input miRNAs. We selected only those target genes which were given by at least two of these databases and excluded those target genes which were validated by only one of these databases. Based on this selection criterion, we obtained a total of 4186 target genes for the top 12 DEMs. Figure 2 is the Venn diagram representation of the results given by these four databases, for example, the value “152” shown in green represents the number of target genes given by both mirDIP and miRWalk (Figure 2).

### 3.4. Construction of the MiRNA–mRNA Network

The construction of miRNA-mRNA network using overlapped genes (target genes vs. CTD (Comparative Toxicogenomics Database) NSCLC genes) was built from SIF files. The up and down regulated network were separately built by Cytoscape as shown in the Appendix A respectively. The upregulated miRNA-target gene interaction network contained 1728 nodes and 1928 edges, wherein, triangles (green) represented the upregulated miRNAs and circles (blue) represented the interacting gene partners. The downregulated miRNA-target gene interaction network contained 1895 nodes and 2256 edges, wherein, diamonds (red) represented the downregulated miRNAs and circles (blue) represented the interacting gene partners. In both upregulated and downregulated miRNA-mRNA network, the target genes were obtained from different databases and the common ones proceeded further.

The merge interaction network constructed using upregulated and downregulated DEMs is represented in Figure 3. The merged network was constructed using Cytoscape software. Further, this built merged network was used for analysis of modules detection. This is the new way to construct the miRNAs-mRNAs network by using SIF Files. If the network was constructed using web tools like miRNet, Network Analyst and MIENTURNET, than all key miRNAs would not have been interacted. In this regard, to find target genes for each miRNA we utilized four different databases (to validate our results, different databases were used to cross-check the target genes). Thus, after obtaining all the overlapped target genes (from four databases), these were further used for construction of the network.

### 3.5. Detection of Significant Modules

Cytoscape software (version 0.1) was explored to detect significant modules as well as top-ranked genes in the miRNA-mRNA network (Figure 4). To reduce the intricacy and intrusion of the unrelated genes from the obtained list of NSCLC genes through regulatory network, some common elements (genes and miRNAs) were identified based on centrality measures, i.e., degree (30 nodes (gene/miRNA) and 92 edges), betweenness (30 nodes (gene/miRNA) and 88 edges), closeness (30 nodes (gene/miRNA) and 86 edges) and stress (30 nodes (gene/miRNA) and 90 edges) (Figure 4). The obtained 13 common elements included nine genes: *CPEB* (Cytoplasmic polyadenylation element binding protein), *SAMD8* (Sterile α motif domain containing 8), *FOXP1* (Forkhead box protein P1), *TRPS1* (Tricho-rhino-phalangeal syndrome 1), *TCF4* (T-cell factor 4), *TBL1XR1* (Transducin (β)-like 1X related protein 1), *SPRED1* (Sprouty-related, *EVH1* domain-containing protein 1), *CELF2* (CUGBP Elav-like family member 2) and *CDK19* (Cyclin-dependent kinase 19); and fourmiRNAs: miR-30a-3p, miR-130b-3p, miR-200b-3p, miR-205-3p. These common elements were referred to as key genes or key miRNAs that were the resultant of significant modules (degree, closeness, betweenness and stress). As an addition to this, the top ranked thirty genes top (having highest degree, closeness, betweenness and stress) were also scrutinized. The Figure 5 is the Venn diagram depiction of the intersection of the top 30 genes in each centrality measure.

### 3.6. Analysis of Gene Term Enrichment and Pathways

Further, the identified top ranked 10 DEMs were systematically characterized to explore their functions and pathways. The DEMs were classified into BP, CC and MF. The GO functional annotation of the 10 candidate NSCLC DEMs biomarkers (4 upregulated and 6 downregulated miRNAs) is represented as heat map (Figure 6). The significant GO categories related to top 10 DEMs included ion binding (MF), RNA binding (MF), cytosol (CC), nucleoplasm (CC), transcription factor activity (MF), biosynthetic process (BP), cell cycle (BP) and signaling pathways (BP).

Additionally, Figure 7 illustrates the enriched pathways of top 10 DEMs (4 upregulated and 6 downregulated miRNAs) associated with NSCLC in the form of generated heat map. The significant signal pathways of DEMs were mainly enriched with Hepatitis B, cell cycle, FoxO signaling pathway and Hippo signaling pathway.

Further, KEGG pathways of the key miRNAs (miR-30a-3p, miR-130b-3p, miR-200b-3p, miR-205-3p) were explored from mieunturnet (Figure 8). The disease ontology of the key miRNAs is shown in the lower panel in Figure 8.

Furthermore, the gene term enrichment analysis was performed to explore the functions and pathways of key genes regulated by DEMs. In the molecular function group, the key genes associated with DEMs were principally involved in DNA binding, ion binding, DNA-binding transcription factor activity, transcription factor binding, RNA binding, and mRNA binding (Figure 9). In the biological process group, the key genes associated with DEMs were linked with anatomical structure development, protein-containing complex assembly, cell differentiation, biosynthetic process, cellular nitrogen compound metabolic process, cellular component assembly, and cellular protein modification process, and lipid metabolic process (Figure 9). In the cellular component group, the key genes associated with DEMs were mainly related to the nucleus, protein-containing complex, cytoplasm, nucleoplasm, plasma membrane, and cytosol (Figure 9).

### 3.7. Survival Plot Analysis of the Key Genes

Survival analysis of obtained key genes was undertaken using GEPIA. The overall survival analysis of the obtained key genes (*CPEB3*, *SAMD8*, *FOXP1*, *TRPS1*, *TCF4*, *TBL1XR1*, *SPRED1*, *CELF2*, and *CDK19*) was examined to link their correlation with the prognosis of NSCLC (Figure 10). Survival curves are used to show the survival ability with time and survival rate (using *p*-value 0.05).Moreover, the GEPIA tool was utilized to validate the expression of key genes between control and lung cancer tissues (in LUSC cohort). It was determined that the miRNA expression of genes *CDK19*, *SAMD8*, *TBL1XR1*, and *TRPS1* were significantly upregulated in the LUSC dataset between lung cancer patients and controls (Figure 11). Moreover, the relation between key gene expression and pathological/tumor stage in NSCLC patients was estimated that revealed the association of key genes with tumor stage NSCLC patients (Figure 12).

### 3.8. Identification of the TFs

The aforementioned key genes were further explored to obtain their TFs. The Enrichr database used inbuild source TRUST, which identified potential TF. Initially, the extracted TF belonged to two organisms, i.e., human and mouse. However, the mouse-associated TFs were excluded from the analysis, and the human-associated TFs are mentioned in the table (Table 3). It was revealed that *TP63* (transformation-related protein 63), *VHL* (von Hippel-Lindau tumor suppressor, *LEF1* (Lymphoid enhancer-binding factor 1), *RUNX3* (Runt-related transcription factor 3), *ESR1* (Estrogen receptor 1), *EGR1* (Early growth response protein 1) and *AR* (Androgen receptor) possibly played significant roles in NSCLC.

## 4. Discussion

Lung cancer has the highest mortality rate among all forms of cancers across the globe. The underlying molecular mechanisms leading to the occurrence and development of NSCLC remain unexplored. Delayed diagnosis and poor prognosis of NSCLC is still a concern, which requires urgent attention. In this context, an in-depth investigation into the mechanisms and factors leading to NSCLC progression is necessary for effective treatment. Common genetic alterations in the development of a particular disease can easily be determined through well-developed microarray technology. It allows the identification of gene targets for appropriate diagnosis, therapy, and prognosis of tumors. Microarray data analyzed using bioinformatics methods, could be utilized for screening cancer biomarkers and therapeutic targets [33]. Kentaro Inamura and Yuichi Ishikawa reported the two main characteristics attributed to miRNAs, due to which they are employed in diagnostics and prognostics as well as the targeted therapeutics in tumors [8], which includes; easy accessibility of miRNAs towards a non-invasive liquid biopsy and the stability of miRNAs in FFPE (formalin-fixed paraffin-embedded) samples. These two features make miRNAs promising biomarkers in cancer diagnosis which could be applied to histological classification or genetic alterations [8].

Though many studies have focused on the relationship between miRNAs and NSCLC, however, a study showing a comparative analysis of the aforementioned miRNA expression profiles (GSE25508, GSE19945, and GSE53882) has not been conducted. Thus, our identified hub miRNAs and their associated genes show a discrepancy with the results obtained in previous studies due to heterogeneity in NSCLC cases and control subjects. For instance, researchers have shown miR-582-5p [34] and miR-107 [35] as prognostic biomarkers which included a total of 30 [34] and 137 [35] matched NSCLC tissue samples and adjacent normal (noncancerous) tissue samples respectively. A recent investigation has also been conducted on eight NSCLC-associated datasets (GSE19188, GSE118370, GSE10072, GSE101929, GSE7670, GSE33532, GSE31547, and GSE31210), however, it identified DEGs and its related target genes [36]. Further, some of the earlier reports have also shown the correlation of miRs to lung cancer as well as specifically NSCLC but these studies comprised different datasets [8,9,10,11,12,13,14,15,16,17,18,19]. Moreover, a bioinformatics integrative analysis carried out on NSCLC by Shao and colleagues, in the year 2017, included only two datasets GSE63459 and GSE36681, which identified some novel miRs as potential biomarkers [37]. Furthermore, some studies have shown the analysis on NSCLC by constructing the circRNA–miRNA–mRNA network (circRNA: circular RNA) [38,39] or lncRNA–miRNA–mRNA network (lncRNA: long non-coding RNA) [40], LINC00973-miRNA-mRNA ceRNA (ceRNA: competing endogenous RNA) [36] but included different datasets altogether (GSE101684 andGSE112214 [38]; GSE102286, GSE112214 and GSE101929 [39]; GSE193628 [40]; GSE27262, GSE89039, GSE101929, GSE40791 and GSE33532 [41]), besides the given datasets used in our present investigation. Some more published literatures on NSCLC included data patient samples like [42,43]. Taking together these findings, it is noteworthy to mention that the present study performed an integrated analysis on some of the other unexplored miRNA expression profiles of NSCLC. In this context, our identified hub miRNAs and their related genes could be implicated in the development and progression of NSCLC. Moreover, since our study performed the bioinformatics analysis of the unexplored miRNA expression profiles, thus, from the initial long lists of miRNAs, we have provided only a few important key miRNAs that can be targeted as therapeutic targets as all miRNAs and their associated target genes cannot form therapeutic targets for the cure of NSCLC.

The study involved three distinguishing microarray expression profiles (GSE25508, GSE19945, and GSE53882). These expression series consisting of tissue samples (NSCLC and normal) were analyzed for the selection of DEMs. Our analysis revealed a total of 172 identified DEMs in NSCLC, which included 82 upregulated and 90 downregulated miRNAs. As a result, 12 overlapping differentially expressed miRNAs were obtained which included 5 upregulated (miR-130b, miR-96, miR-210, miR-200b, and miR-205) and 7 downregulated (miR-30a, miR-145, miR-140-3p, miR-572, miR-144, miR-126, and miR-486-5p) miRNAs. After a literature search, it was found that these obtained DEMs have been involved in lung cancer expression profiling studies [11]. The upregulated miRNAs revealed in our study (miR-130b, miR-96, miR-210, miR-200b and miR-205) show consistency with the previous profiling studies (between normal tissue samples and lung cancer samples), that has reported the 26 consistently up-regulated miRNAs (miR-210, miR-21, miR-182, miR-31, miR-205, miR-200b, miR-183, miR-203, miR-196a, miR-708, miR-92b, miR-193b, miR-106a, miR-135b, miR-96, miR-17-5p, miR-20b, miR-18a, miR-200a, miR-93, miR-130b, miR-200c, miR-375, miR-20a, miR-18b) [11]. MiR-130b, was found to be the miRNA that showed the largest (most significant) deviation in lung cancer patients in those that developed metastases from controls (*p* = 0.0004) as compared to those patients that did not develop metastases [44]. MiR-130b (onco-miRNA) is straightaway regulated by NF-*κ*B and assists in NF-*κ*B activation by lessening cylindromatosis expression. MiR-130b-3p also downregulates PTEN expression and promotes the proliferation, invasion, migration, and cytoskeletal rearrangement by activating PI3K and integrin *β*1signaling pathways. Moreover, the inhibitors of miR-130b-3p have been shown to induce apoptosis [45,46,47]. Mir-96 (member of family miR-183) up-regulation has been reported in breast cancer [48]. Reports have suggested that miR-96 performs both oncosuppressor and oncogene functions by lessening or promoting cell growth in different cancer types [49,50,51,52]. The significant increase of miR-96 expression in our NSCLC miRNA profiling, as compared with normal tissue, is in accordance with published reports [53,54,55]. A study has reported the up-regulation of miR-96 by targeting FOXO3 as its major gene [56]. The expression of RAD51 and REV1 (related to DNA repair and homologous recombination) is downregulated by miR-96 that perhaps has a vital role in DNA repair inhibition and chemosensitivity [57]. MiR-210 has been previously linked to lung cancer through the modulation of the JAK2/STAT3 pathway [58]. A recent study (on lung cancer) has demonstrated the role of miR200b as a possible biomarker for PD-L1 expression [59]. The expression of miR200b is inversely proportional to PD-L1 expression, i.e., low miR200b expression was related to High PD-L1 expression, whereas high miR200b expression was linked to low PD-L1 expression in human lung cancer patients [59]. MiR-205 has been recognized as an extremely accurate biomarker for lung cancer (squamous) [60]. MiR-205 has been revealed as a biomarker in lung squamous cell carcinoma specifically [61].

Addtionally, our identified six downregulated miRNAs (miR 30a, miR 145, miR 140 3p, miR 572, miR 144, miR 126 and miR 486 5p) are also in agreement with the previous investigation, that has reported consistently (a total of 28) down-regulated miRNAs in profiling studies (miR-126, miR-30a, miR-451, miR-486-5p, miR-30d, miR-145, miR-143, miR-139-5p, miR-126, miR-140-3p, miR-138, miR-30b, miR-486, miR-101, miR-125a, miR-198, miR-144, miR-140, miR-218, miR-32, miR-338-3p, miR-99a, miR-195, miR-497, miR-30c, miR-130a, miR-16, miR-139) [11]. The downregulation of miR-30a has been recently reported in a recent investigation carried out in NSCLC using a computational approach [51]. It has been suggested that mir-30a expression could have an effect on the NSCLC patient’s survival rates [62]. MiR-126 and miR-145 have been documented to perform roles in NSCLC by inhibiting the growth of tumor growth cells [63]. MiR-126 has been shown to enhance the sensitivity of NSCLC cells to an anticancer agent by targeting VEGFA (vascular endothelial growth factor A) [64]. MiR-140-3p has been suggested as a biomarker in squamous cell carcinoma [65]. Lung cancer-related microarray analysis has shown miR-144 as one of the most significantly down-regulated miRNAs [66]. The downregulated expression of miR-486-5p is in accordance with a recently published study, which has shown the inhibition of NSCLC through mTOR signaling pathway repression via targeting ribosomal proteins [67]. Moreover, the study demonstrated miR-486-5p as a promising biomarker during the early stages of NSCLC [67]. The consistently reported upregulated miRNA miR-210 has been reported in nine studies (average FC: 2.65) while miR-21 has been reported in seven studies (average FC: 4.39). Also, consistently reported downregulated miRNA miR-126 has been reported in ten studies (average FC: 0.33) while miR-30a has been reported in eight studies (average FC: 0.36). Also, studies have shown miR-210 as the most frequently reported upregulated miRNA in both squamous carcinoma-based analysis [65,68,69,70,71,72] and adenocarcinoma [73,74,75]. These investigations are in line with our present hypothesis.

By incorporating the centrality concept [76] and its associated algorithms we constructed the miRNA–mRNA interaction network through analyzing the topological properties (degree, betweenness, closeness, and stress) to expose the key miRNAs regulating the network. MiR-30a-3p, miR-130b-3p, miR-200b-3p and miR-205-3pwere identified as 4 key miRNAs [76]. Among these aforementioned key miRNAs miR-130b-3p, miR-200b-four and miR-205-3p were found to be upregulated while miR-30a-3p showed downregulated expression in NSCLC. As already mentioned, these miRNAs have been previously reported, thus showing consistency with earlier published literatures. This suggested a significant role of these miRNAs played in the regulation of NSCLC. However, it is interesting to mention that the reported downregulated expression of miR-572 is exclusive to our present study.

Further, key genes regulating the network were identified from the constructed miRNA–mRNA network that included *CPEB3*, *SAMD8*, *FOXP1*, *TRPS1*, *TCF4*, *TBL1XR1*, *SPRED1*, *CELF2*, and *CDK19.* CPEB (cytoplasmic polyadenylation element-binding protein) is an RNA-binding protein which interacts with CPE (cytoplasmic polyadenylation element) in the 3′UTRs of specific mRNAs to repress or activate translation [77]. The role of *CPEB3* (as a tumor suppressor) has recently been demonstrated in colorectal cancer through regulation (post-transcriptional) of the JAK/STAT signaling pathway [78]. Its downregulated expression has also been reported in cervical cancer [79] and human HCC [80]. A study has examined the role of FOXP1 (Forkhead box protein P1) in lung cancer which suggested its essentiality in preventing the development of lung adenocarcinoma via suppression of chemokine signaling pathways [81]. It is also considered a potential therapeutic target in cancer [82,83]. The report has shown the association of TRPS1 (Tricho-rhino-phalangeal syndrome 1) with MDR (multidrug resistance) in lung cancer [84,85,86]. Trps1 (GATA protein) has been shown as a potential tumor marker (cytologic) in a variety of cancers (osteosarcoma, malignant tumor, prostatic carcinoma, and breast cancer) [87,88,89] and plays an imperative role in the differentiation and enlargement of mammals [90,91]. TCF-4 (T cell factor-4) carries out important roles in the development and carcinogenesis. High expression of TCF-4 was revealed in NSCLC samples as compared to normal tissues [92]. However, later a report showed an association between NLK (Nemo-like kinase), a member of the protein kinase (serine/threonine) superfamily, and TCF4 (T-cell factor 4), a transcription factor substrate for NLK in case of lung cancer that revealed that NLK expression was found to be negatively correlated with the expression of TCF4 in lung cancer advancement [93]. TCF-4, as a component of the Wnt pathway, also works as a tumor suppressor in colon cancer [94] and is involved in papillary thyroid carcinoma via regulation of HCP5 [95]. TBL1XR1 (Transducin (β)-like 1X related protein 1) is a subunit of SMRT/NCoR repressor complexes and is essentially required for activating signaling pathways [96]. TBL1XR1 is recognized as the prognostic marker of NSCLC and is found to be related to gastric, breast, and stomach cancers [97]. SPRED1 (Sprouty-related, EVH1 domain-containing protein 1) has been reported as a tumor repressor in paediatric acute myeloblastic leukaemia [98]. CELF2 (CUGBP Elav-like family member 2), an RNA binding protein isoform of CELF, performs important functions in the development and activation of T cells [99] CELF2 acts as a tumor suppressor for a variety of cancers, (ovarian cancer, breast cancer, gastric cancer, glioma, hepatocellular carcinoma, including lung cancer and thus is considered as a biomarker in lung squamous cell carcinoma and breast cancer [100,101]. It has been documented that the growth of NSCLC cells could be suppressed by CELF2 via inhibition of the PREX2-PTEN interaction [102].) CDKs (Cyclin-dependent kinases) play role in many critical processes, such as cell cycle, communication, transcription, metabolism, and apoptosis [103]. Not much is known about the CDK19 mechanism regarding their mediator kinase functions [104]. These key genes were proceeded for enrichment analysis. Furthermore, the GO analysis was performed to explore the biological function of genes regulated by key miRNAs. The GO analysis revealed that hub genes mainly participated in biosynthetic process (BP), anatomical structure development (BP) lipid metabolic process (BP), cell differentiation (BP), nucleoplasm (CC), plasma membrane (CC), nucleus (CC), cytoplasm (CC), DNA binding, ion binding, DNA-binding transcription factor activity (MF), transcription factor binding (MF), RNA binding (MF) and mRNA binding (MF). Thus, these predicted functions further substantiate our findings.

Moreover, the expression plots of the identified key genes showed a significant correlation with NSCLC prognosis. On combining these results, it could be interpreted that the key genes played a significant role in the regulation of NSCLC. For further elucidation, the key genes were predicted for their TFs. *TP63*, a member of tumor suppressor protein p53 [105], is known to associate with the development and tumorigenesis of cancers, in particular with cancer metastasis [106]. VHL is a product of the tumor suppressor gene *VHL*. Recently, a study on human kidney cells has shown its function in cell growth regulation and differentiation [107]. LEF1, a protein encoded by the *LEF1* gene in humans, has shown its expression in several cancers [108]. This protein belongs to TCF (T-cell Factor) family, thus is involved in the Wnt signaling pathway and is vital for stem cell maintenance and organ development [109]. RUNX3, a protein encoded by the *RUNX3* gene in humans and a component of TGF-β (transforming growth factor-β), has shown tumor-suppressive effects in several cancers [110,111]. ESR1, a protein encoded by the *ESRI* gene, has shown its association with many kinds of cancers (endometrial, breast, and prostate) [112]. EGR1 is chiefly involved in tissue injury, fibrosis, and immune response processes. Recent reports have shown the involvement of EGR1 in the initiation and succession of cancer. Nevertheless, the precise mechanism of EGR1 modulation remains unexplored [113]. AR (Androgen receptor), a ligand-dependent transcription factor, has been shown to involve in prostate cancer [114]. Thus, it could be interpreted that these identified TFs may play a significant role in the pathogenesis of NSCLC.

Interestingly, our observations have shown that besides NSCLC other literature have also documented the significance of these aforementioned key genes/miRNAs in other types of cancers. Therefore, these signature genes/miRNAs can largely offer benefits as biomarkers to other cancers as well, besides NSCLC, in diagnosis. Similar reports have also demonstrated the significance of key miRNAs and associated target genes in various syndromes [115,116,117]. It is noteworthy to mention that *SAMD8* is exclusive to our study like miR-572. The utilization of the aforementioned NSCLC biomarkers (identified in our study) could lead to earlier diagnosis resulting in efficient treatment of lung cancer and reduction in disease occurrence, and overall better chances of survival and ultimately a better life for patients with lung cancer. The revealed TFs have also been shown in other cancers. The study presented here also comprises some limitations such as, the inclusion of a comparatively smaller sample size and lack of experimental validations. To validate our findings the identified key miRNAs/gene should be further investigated in a larger number of patients through experiments.

## 5. Conclusions

This study used a bioinformatics approach to analyze the miRNA expression profiles consisting of NSCLC tissue samples and adjacent normal tissue samples. Though previous reports have shown the correlation of signature miRNAs with NSCLC [12,13,14,15], however, a comparative meta-analysis on the retrieved miRNA profiles has not been conducted on NSCLC specifically. Our results identified 12 overlapping miRNAs that were differentially expressed among three expression series, out of which 5 were upregulated and 7 were downregulated. The centrality-based method was employed which revealed four key miRNAs and nine genes. We performed the GO analysis of the key genes/miRNAs which were shown to participate in the biosynthetic process, nucleoplasm, DNA binding, ion binding, RNA binding, and signaling pathways. Further, we carried out the survival, expression, and pathological analyses of the identified hub genes using GEPIA. Subsequently, the TFs were predicted for the key genes in humans. These identified genes/miRNAs can serve as a potential prognostic predictor of patients with NSCLC. However, these signature genes/miRNAs warrant further investigations due to lack of experimental evidence. Therefore, these obtained results from this bioinformatics analysis require validation through experimental research, such as qRT-PCR and Western Blot, to understand the molecular mechanisms of NSCLC.

## Figures and Tables

**Figure 1 genes-13-01174-f001:**
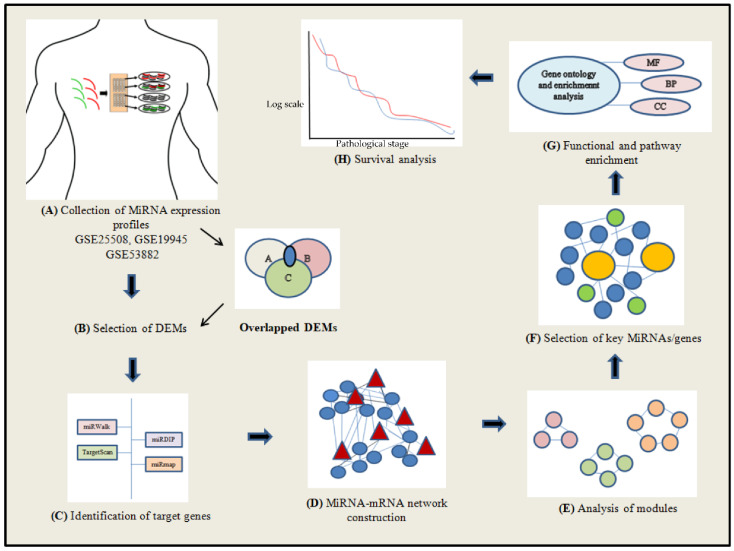
Illustration of the network-based integrative method used in the study. (**A**) Three NSCLC series (GSE25508, GSE19945, and GSE53882) were used for the present analysis. (**B**) The DEMs were identified using comparative approach. (**C**) The target genes associated with DEMs were identified. (**D**) The mRNA–miRNA network was constructed. (**E**) The significant modules based on centrality methods were detected. (**F**) The key miRNAs and their associated genes were obtained. (**G**) Enrichment of function and pathway analysis was performed for the identified key elements (key miRNAs and key genes). (**H**) Survival analysis for the obtained key genes was conducted through survival plots.

**Figure 2 genes-13-01174-f002:**
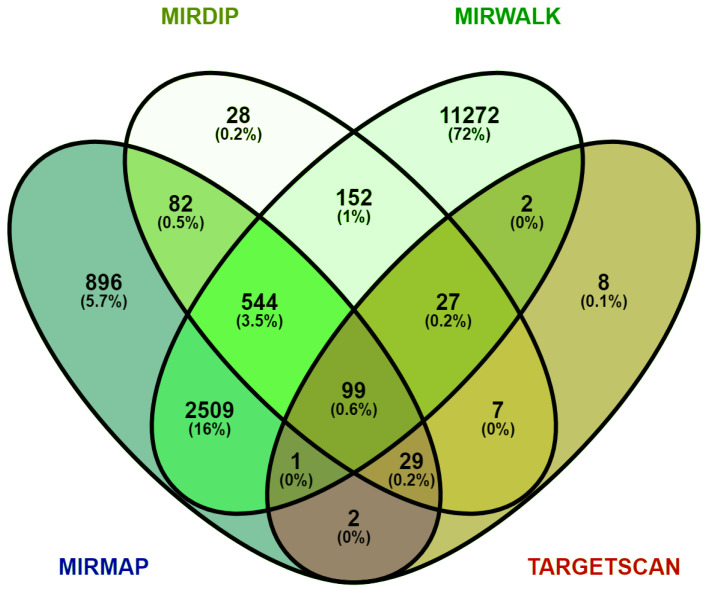
Venn diagram showing overlapping genes between four target predictive databases, i.e., TargetScan, miRWalk, mirDIP, and miRmap. For each miRNA, target genes were retrieved using these four databases; each database showed some different target genes, but we extracted the common genes that were validated in all databases. Abbreviations: MiRNAs: MicroRNAs.

**Figure 3 genes-13-01174-f003:**
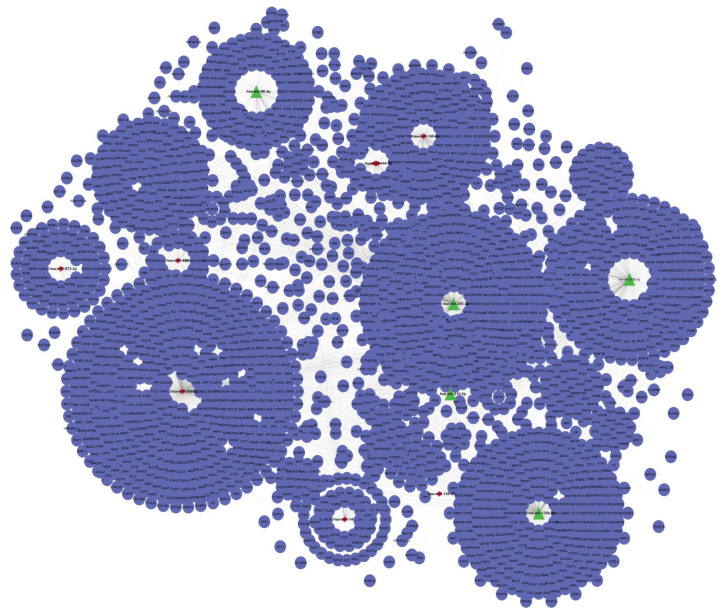
MiRNA-mRNA interaction merge network. The interaction network is constructed with the upregulated and downregulated DEMs using Cytoscape software. The miRNA-mRNA consists of 2970 nodes and 4184 edges. Triangle (green) represents upregulated miRNAs, diamond (red) represents downregulated miRNAs, and circle (blue) represents the interacting partners.

**Figure 4 genes-13-01174-f004:**
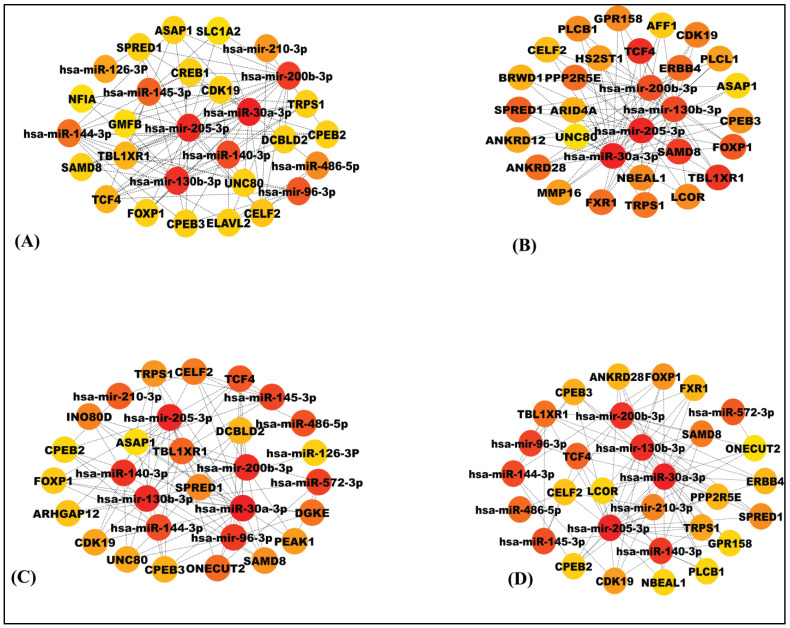
Significant modules and the top 30 ranked genes/miRNAs in the network on the basis of (**A**) Degree, (**B**) Closeness, (**C**) Betweenness and (**D**) Stress. The top 30 ranked genes/miRNAs in the network indicate both gene and miRNA. Red indicates the highest rank, whereas yellow indicates the lowest rank. Based on these modules, nine key genes and four miRNAs were found to be common in all modules and were considered as significant hub nodes.

**Figure 5 genes-13-01174-f005:**
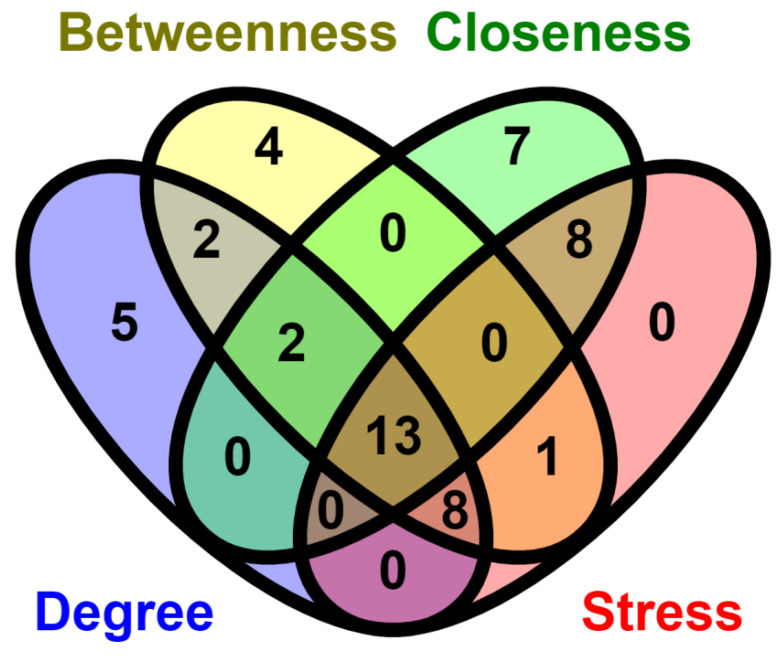
Venn diagram depicting the overlapped nodes in four methods used in Cytohubba. Showing intersections of topological properties.

**Figure 6 genes-13-01174-f006:**
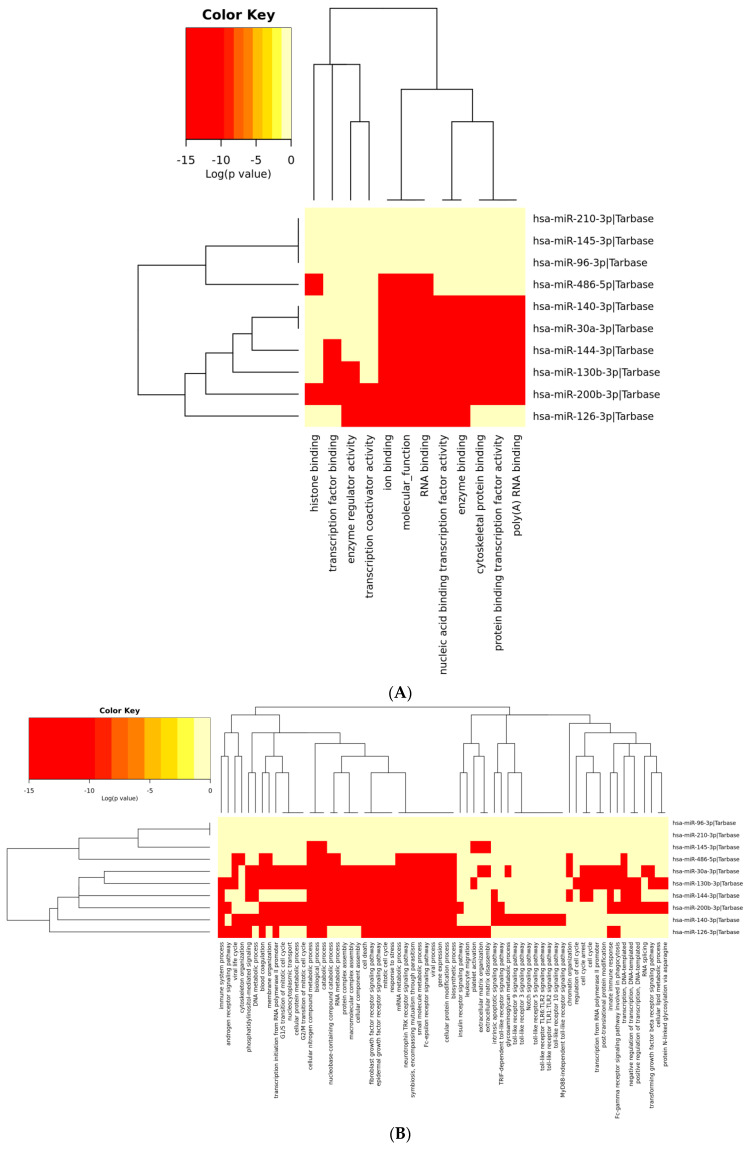
Gene Ontology of the 10 candidate sepsis DEM biomarkers. Representation of functional enrichment of miRNAs showing (**A**) miRNAs versus Molecular Functions, (**B**) miRNAs versus Biological Processes, and (**C**) miRNAs versus Cellular components. Abbreviations: DEMs: Differentially expressed miRNAs.

**Figure 7 genes-13-01174-f007:**
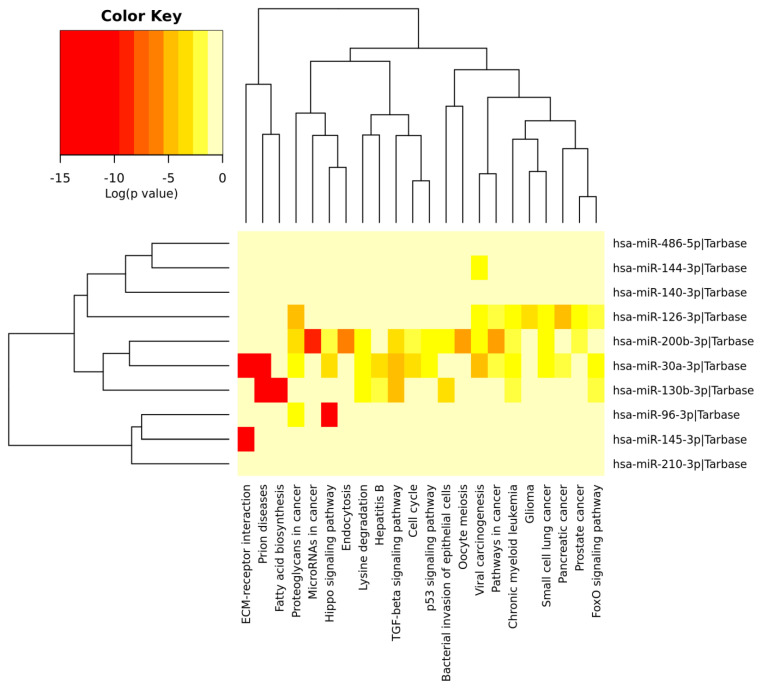
Heatmap of the 10 DEMs (4 upregulated and 6 downregulated miRNAs) showing the top 10 enriched functional pathways (*X*₋axis represents the name of the pathways and *Y*₋axis represents the miRNAs).

**Figure 8 genes-13-01174-f008:**
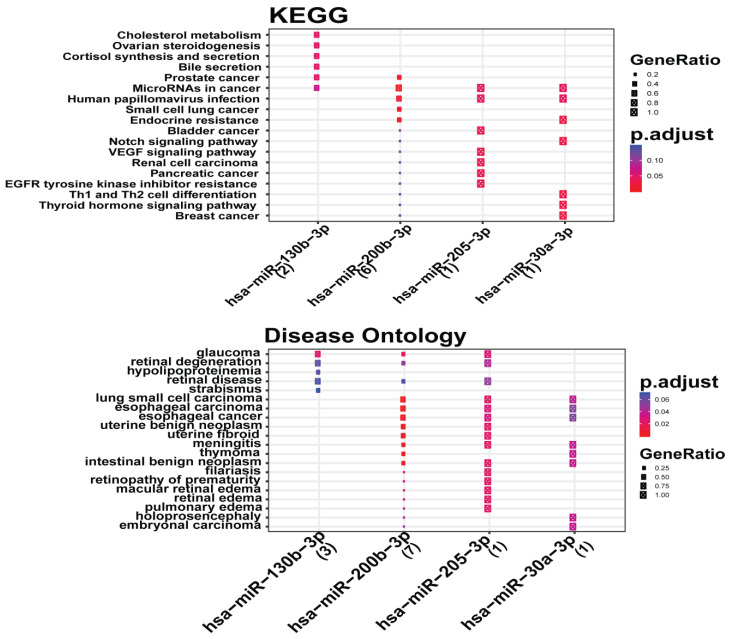
MiRNA enrichment analysis. (Upper Panel) Mieunturnet used to explore the KEGG pathways of miRNAs. (Lower Panel) Disease ontology of miRNAs. Abbreviations: KEGG: Kyoto Encyclopedia of Genes and Genomes.

**Figure 9 genes-13-01174-f009:**
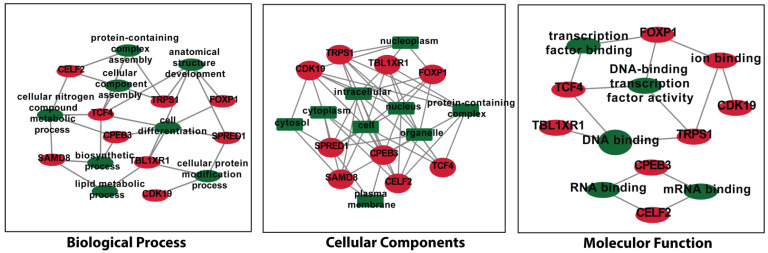
GO of the NSCLC associated key genes (*CPEB3*, *SAMD8*, *FOXP1*, *TRPS1*, *TCF4*, *TBL1XR1*, *SPRED1*, *CELF2,* and *CDK19*) from the miRNA-mRNA network as a result of four significant modules. Abbreviations: *CPEB*: Cytoplasmic polyadenylation element binding protein; *SAMD8*: Sterile α motif domain containing 8; *FOXP1*: Forkhead box protein P1; *TRPS1*: Tricho-rhino-phalangeal syndrome 1; *TCF4*: T-cell factor 4; *TBL1XR1*: Transducin (β)-like 1X related protein 1; *SPRED1*: Sprouty-related, EVH1 domain-containing protein 1; *CELF2*: CUGBP Elav-like family member 2; *CDK19*: Cyclin-dependent kinase 19.

**Figure 10 genes-13-01174-f010:**
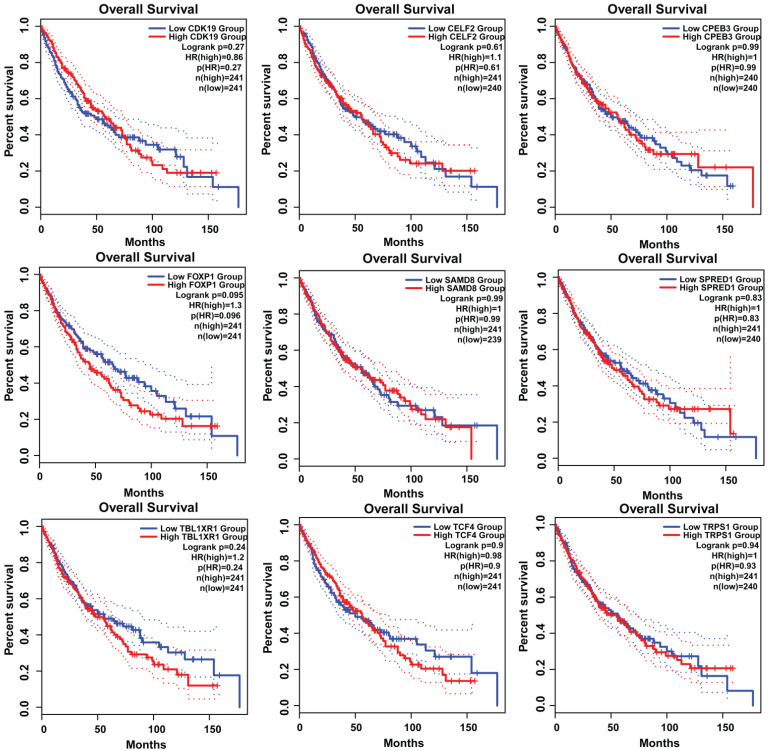
Survival analysis of key genes *CPEB3*, *SAMD8*, *FOXP1*, *TRPS1*, *TCF4*, *TBL1XR1*, *SPRED1*, *CELF2,* and *CDK19*. The survival curves of key genes in patients with NSCLC were obtained from GEPIA. Survival plots were used to show the survival ability with time and survival rate.

**Figure 11 genes-13-01174-f011:**
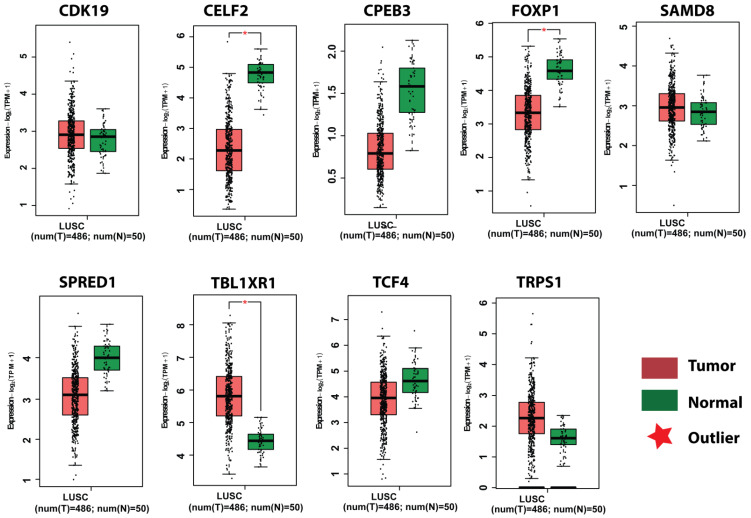
Expression analysis of selected key genes *CPEB3*, *SAMD8*, *FOXP1*, *TRPS1*, *TCF4*, *TBL1XR1*, *SPRED1*, *CELF2*, and *CDK19*. Box plots obtained from GEPIA showing expression profiles of key genes in tumor (red) and normal (green) samples of LUSC datasets (*p* < 0.05) in patients with NSCLC. ‘Outlier’ represents the statistical difference in gene expression between two boxplots.

**Figure 12 genes-13-01174-f012:**
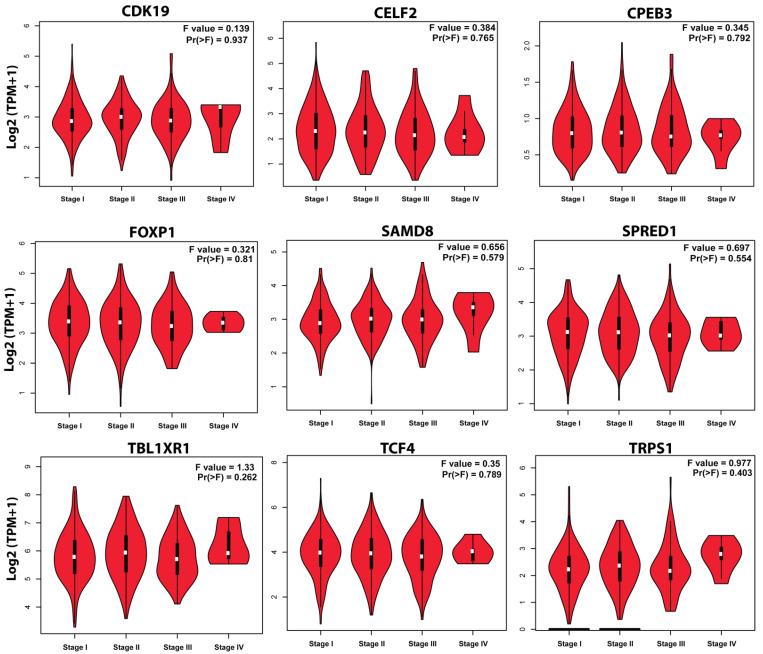
Pathological analysis of selected key genes *CPEB3*, *SAMD8*, *FOXP1*, *TRPS1*, *TCF4*, *TBL1XR1*, *SPRED1*, *CELF2*, and *CDK19* in NSCLC. Violin plots obtained from GEPIA showing pathological stage condition for key genes using LUSC datasets in patients with NSCLC. The *X*₋axis represents the major pathological stages while the *Y*₋axis represents the log scale transformed expression data.

**Table 1 genes-13-01174-t001:** List of datasets used in the network analysis.

Series	TS	N	D	UR	DR	GPL	C	Y
GSE25508	60	26	34	39	52	7731	Finland	2011
GSE19945	63	8	55	14	31	9948	Japan	2013
GSE53882	548	397	151	29	7	18130	China	2017

TS: Total samples; N: Normal; D: Disease; UR: Upregulated; DR: Downregulated; Country; Y: Year.

**Table 2 genes-13-01174-t002:** Top 12 DEMs (7 downregulated and 5 upregulated) on the basis of log fold change and *p*-value. Upregulated miRs are miR-210, miR-130b, miR-96, miR-200b and miR-205 and downregulated miRs are miR-30a, miR-145, miR-140 3p, miR-572, miR-144, miR-126 and miR-486-5p.

Adjusted *p*-Value	*p*-Value	Log FC	miRNA	OG	OG vs. CTD
0.000488	3.6 × 10^−7^	3.44103	MiR-30a	1076	1050
0.002008	4.63 × 10^−6^	4.13116	MiR-145	154	149
0.002008	4.59 × 10^−6^	1.95769	MiR-140-3p	387	370
0.002008	6.86 × 10^−6^	2.22649	MiR-572	122	118
0.002008	7.9 × 10^−6^	1.68913	MiR-144	144	137
0.004767	3.24 × 10^−5^	2.08135	MiR-126	11	10
0.008815	1.93 × 10^−4^	2.66117	MiR-486-5p	99	95
0.014676	1.13 × 10^−3^	−1.71609	MiR-210	26	26
0.014796	1.17 × 10^−3^	−1.88279	MiR-130b	592	586
0.018155	2.38 × 10^−3^	−1.65166	MiR-96	290	285
0.004867	4.90 × 10^−6^	−0.71609	MiR-200b	573	560
0.006767	7.90 × 10^−6^	−0.65166	MiR-205	832	800

Log FC: Log fold change; OG: Overlapped genes; CTD: Comparative Toxicogenomics Database. The overlapped genes were predicted by four databases: TargetScan, miRWalk, mirDIP, and miRmap.

**Table 3 genes-13-01174-t003:** Transcription factors prediction through TRUST.

Term	*p*-Value	Adjusted *p*-Value	Key Genes
TP63 human	0.006731046	0.029080631	TCF4
VHL human	0.008965773	0.029080631	TCF4
LEF1 human	0.012977027	0.029080631	TCF4
RUNX3 human	0.013421829	0.029080631	TCF4
ESR1 human	0.033691182	0.044511811	FOXP1
EGR1 human	0.038917584	0.044511811	TCF4
AR human	0.041087826	0.044511811	TRPS1

Abbreviations: *TP63*: Transformation-related protein 63; *VHL*: Von Hippel-Lindau tumor suppressor; *LEF1*: Lymphoid enhancer-binding factor 1; *RUNX3*: Runt-related transcription factor 3; *ESR1*: Estrogen receptor 1; *EGR1*: Early growth response protein 1; *AR*: Androgen receptor.

## Data Availability

Publicly available GEO database (https://www.ncbi.nlm.nih.gov/geo/, accessed on 1 June 2022).

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
