# Peer review of "Identification of the Key miRNAs and Genes Associated with the Regulation of Non-Small Cell Lung Cancer: A Network-Based Approach"

_genes, 2022, doi:10.3390/genes13071174_

Round 1

Reviewer 1 Report

The authors performed bioinformatic analysis using three published miRNA expression datasets to identify the key miRNAs and their potential target genes that are involved in the non-small cell lung cancer (NSCLC). NSCLC associated miRNA-mRNA network, survival analysis and transcription factors were also provided. However, some issues in the manuscript are needed to be addressed.

1.    Many studies focusing on NSCLC associated miRNA-mRNA network, survival analysis have been published. I do not gain much new information from this study as (1) the most of the key miRNAs and target genes identified in this study were known to play roles in NSCLC or the other cancer types, which was also mentioned in the discussion of the manuscript; (2) the authors did not develop or provide new bioinformatic approaches to identify the NSCLC associated miRNAs and target genes and their interaction network. I suggest the authors should emphasize the novelty in their manuscript.

2.    Table 2. All the up-regulated miRs were listed as precursor miRNAs (mir-), whereas the down-regulated miRNAs were listed as mature miRNAs (miR-). The authors should have a double check with the results to make sure what they identified are mature miRNAs or precursor miRNAs in the manuscript.

3.    line 229. Table 2 should be Table 3. I suggest to remove this table from the manuscript or list them in a supplemental file.

4.    Figure 5 and Figure 8. The quality of figure should be improved.

5.    line 330. The conclusion “The survival curves of the key miRNAs and genes were found to be significantly associated with NSCLC prognosis” is not supported by the Figure 12. Figure 12 did not show that the expression of target gene was significantly correlated with the survival. P value should be provided. The authors also need to provide the overall survival analysis of the differentially expressed miRNAs in NSCLC patients.

6. The authors identified several key miRNA target genes to be significantly associated with NSCLC, however, some of them are common targets by up-regulated and down-regulated miRNAs, which makes the results are less persuasive. For example, CDK19 is targeted by miR-130b, miR-205, miR-30a, miR-140-3p, miR-144.

7.    Many typos were found in the manuscripts.

Author Response

Author’s point by point response to the Reviewer comments

We have seen the comments and noted the concerns raised by the Editor and Reviewers. In this context we have provided some additional information in the point-by-point response to make it more explicit. We also appreciate the efforts of the Editor and esteemed reviewers whose comments have indeed uplifted the quality of our manuscript.

Reviewers Comments and Author’s Response:

Reviewer 1

Comments and Suggestions for Authors The authors performed bioinformatic analysis using three published miRNA expression datasets to identify the key miRNAs and their potential target genes that are involved in the non-small cell lung cancer (NSCLC). NSCLC associated miRNA-mRNA network, survival analysis and transcription factors were also provided. However, some issues in the manuscript are needed to be addressed:

  1. Many studies focusing on NSCLC associated miRNA-mRNA network, survival analysis have been published. I do not gain much new information from this study as (1) the most of the key miRNAs and target genes identified in this study were known to play roles in NSCLC or the other cancer types, which was also mentioned in the discussion of the manuscript; (2) the authors did not develop or provide new bioinformatics approaches to identify the NSCLC associated miRNAs and target genes and their interaction network. I suggest the authors should emphasize the novelty in their manuscript.

Response 1. We would like to thank the reviewer for providing insightful comments. As suggested, the paragraph discussing the novelty of this study has been incorporated in the revised version of the manuscript.

Introduction Section (Page 2)              

Though these aforementioned studies have documented the role of miRNAs in lung cancer or specifically NSCLC, these reports comprised different datasets [8 - 19]. Therefore, in this regard, our present study performed an integrated analysis on some of the other unexplored gene expression profiles of NSCLC. Thus, our identified key miRNAs and related genes show discrepancy with the previous study results due to heterogeneity in NSCLC cases and control subjects. In this study, we used the network analyses to show the correlation between the identified key miRNAs/genes and NSCLC. This kind of study is envisaged to provide useful information in exploring candidate miRNA biomarkers in human NSCLC.”

Discussion Section (Pages: 28 - 29)

Though many studies have focused on the relationship between miRNAs and NSCLC, however, a study showing comparative analysis on aforementioned miRNA expression profiles (GSE25508, GSE19945 and GSE53882) has not been conducted. Thus, our identified hub miRNAs and its associated genes show discrepancy with the results obtained in previous studies due to heterogeneity in NSCLC cases and control subjects.  For instance, researchers have shown miR-582-5p [29] and miR-107 [30] as prognostic biomarkers which included a total of 30 [29] and 137 [30] matched NSCLC tissue samples and adjacent normal (noncancerous) tissue samples respectively. A recent investigation has also been conducted on 8 NSCLC-associated datasets (GSE19188, GSE118370, GSE10072, GSE101929, GSE7670, GSE33532, GSE31547, and GSE31210), however, it identified DEGs and its related target genes [31]. Further, some of the earlier reports have also shown the correlation of miRs to lung cancer as well as specifically NSCLC but these studies comprised different datasets [8 - 19]. Moreover, a bioinformatics integrative analysis carried out on NSCLC by Shao and colleagues, in the year 2017, included only two datasets GSE63459 and GSE36681, which identified some novel miRs as potential biomarkers [32].   Furthermore, some studies have shown the analysis on NSCLC by constructing the circRNA–miRNA–mRNA network (circRNA: circular RNA) [33, 34] or lncRNA–miRNA–mRNA network  (lncRNA: long non-coding RNA) [35], LINC00973-miRNA-mRNA ceRNA (ceRNA: competing endogenous RNA) [36] but included different datasets altogether (GSE101684 and GSE112214 [33]; GSE102286,  GSE112214 and  GSE101929 [34]; GSE193628 [35]; GSE27262, GSE89039, GSE101929, GSE40791  and GSE33532 [36]), besides the given datasets used in our present investigation. Taken together these findings, it is noteworthy to mention that the present study performed an integrated analysis on some of the other unexplored miRNA expression profiles of NSCLC. In this context, our identified hub miRNAs and its related genes could be implicated in the development and progression of NSCLC. Moreover, since our study performed the bioinformatics analysis of the unexplored miRNA expression profiles, thus, from the initial long lists of miRNAs, we have provided only few important key miRNAs that can be targeted as therapeutic targets as all miRNAs and their associated target genes cannot form therapeutic targets for the cure of NSCLC”.

Further, miR-572 and SAMD8 have been reported for the first time in our study as mentioned in the discussion section.

Page 30

 “However, it is interesting to mention that the reported down-regulated expression of miR-572 is exclusive to our present study”.

Page 31

It is noteworthy to mention that SAMD8 is exclusive to our study like miR-572.

Added References:

  1. Bhattacharyya N, Gupta S, Sharma S, Soni A, Bagabir SA, Bhattacharyya M, Mukherjee A, Almalki AH, Alkhanani MF, Haque S, Ray AK. CDK1 and HSP90AA1 appear as the novel regulatory genes in non-small cell lung cancer: a bioinformatics approach. Journal of personalized medicine. 2022 Mar 4;12(3):393.
  2. Shao Y, Liang B, Long F, Jiang SJ. Diagnostic microRNA biomarker discovery for non-small-cell lung cancer adenocarcinoma by integrative bioinformatics analysis. BioMed research international. 2017 Oct;2017.
  3. Yang J, Hao R, Zhang Y, Deng H, Teng W, Wang Z. Construction of circRNA-miRNA-mRNA network and identification of novel potential biomarkers for non-small cell lung cancer. Cancer cell international. 2021 Dec;21(1):1-7.
  4. Cai X, Lin L, Zhang Q, Wu W, Su A. Bioinformatics analysis of the circRNA–miRNA–mRNA network for non-small cell lung cancer. Journal of International Medical Research. 2020 Jun;48(6):0300060520929167.
  5. Ding D, Zhang J, Luo Z, Wu H, Lin Z, Liang W, Xue X. Analysis of the lncRNA–miRNA–mRNA Network Reveals a Potential Regulatory Mechanism of EGFR-TKI Resistance in NSCLC. Frontiers in genetics. 2022 Jan;13.
  6. Guo Q, Li D, Luo X, Yuan Y, Li T, Liu H, Wang X. The Regulatory Network and Potential Role of LINC00973-miRNA-mRNA ceRNA in the Progression of Non-Small-Cell Lung Cancer. Frontiers in immunology. 2021;12.
  7. Arora S, Singh P, Ahmad S, Ahmad T, Dohare R, Almatroodi SA, Alrumaihi F, Rahmani AH, Syed MA. Comprehensive Integrative Analysis Reveals the Association of KLF4 with Macrophage Infiltration and Polarization in Lung Cancer Microenvironment. Cells. 2021 Aug;10(8):2091.
  8. Peng X, Guan L, Gao B. miRNA-19 promotes non-small-cell lung cancer cell proliferation via inhibiting CBX7 expression. OncoTargets and therapy. 2018;11:8865.

  1. Table 2. All the up-regulated miRs were listed as precursor miRNAs (mir-), whereas the down-regulated miRNAs were listed as mature miRNAs (miR-). The authors should have a double check with the results to make sure what they identified are mature miRNAs or precursor miRNAs in the manuscript.

Response 2. Thank you for the suggestion. All miRNAs are mature form and not precursor. The line emphasizing this have been added in the text as “From these datasets, the 12 overlapped DEMs were identified of which 5 were upregulated and 7 were downregulated (Note: All miRNAs are mature miRNAs) (Page 6).

The corrections have also been made in Table 2 in the column miRNA (Pages: 6 - 7).

 “Upregulated miRs are miR-210, miR-130b, miR-96, miR-200b and miR-205 and downregulated miRs are miR-30a, miR-145, miR-140 3p, miR-572, miR-144, miR-126 and miR-486-5p”.

  1. 3.    line 229. Table 2 should be Table 3. I suggest to remove this table from the manuscript or list them in a supplemental file.

Response 3. As suggested, Table 2 has now been replaced with a supplementary file (Additional File: S2) in the revised version of the manuscript. 

  1. 4.    Figure 5 and Figure 8. The quality of figure should be improved.

Response 4. We thank the reviewer for highlighting this. As suggested, the figures with poor resolution have been replaced with better resolution ones in the revised text (Pages: 11, 15 and 16).  

Note: Figure 5 and Figure 8 now Figure 3 and Figure 6 in the revised text. Moreover, some lines have been added (Pages: 8 - 9).

The construction of miRNA-mRNA network using overlapped genes (target genes vs CTD (Comparative Toxicogenomics Database) NSCLC genes) build from SIF files. The up and down regulated network were separately built by Cytoscape as shown in the additional material (Additional File: S3) and (Additional File: S4) respectively. The upregulated miRNA-target gene interaction network contained 1728 nodes and 1928 edges, wherein, triangles (green) represented the upregulated miRNAs and circles (blue) represented the interacting gene partners. The downregulated miRNA-target gene interaction network contained 1895 nodes and 2256 edges, wherein, diamonds (red) represented the downregulated miRNAs and circles (blue) represented the interacting gene partners. In both upregulated and downregulated miRNA-mRNA network, the target genes were obtained from different databases and the common ones were preceeded forward”.

:The merge interaction network constructed using upregulated and downregulated DEMs is represented in Figure 3. The merged network was constructed using Cytoscape software. Further, this built merged network was used for analysis of modules detection. This is the new way to construct the miRNAs-mRNAs network by using SIF Files. If the network was constructed uisng web tools like miRNet, Network Analyst and MIENTURNET, than all key miRNAs would not have been interacted. In this regard, to find target genes for each miRNA we utilized four different databases (to validate our results, different databases were used to cross-check the target genes). Thus, after obtaining all the overlapped target genes (from 4 databases), these were further used for construction of the network”.

  1. line 330. The conclusion “The survival curves of the key miRNAs and genes were found to be significantly associated with NSCLC prognosis” is not supported by the Figure 12. Figure 12 did not show that the expression of target gene was significantly correlated with the survival. P value should be provided. The authors also need to provide the overall survival analysis of the differentially expressed miRNAs in NSCLC patients.

Response 5. We thank the reviewer for pointing out this mistake. We would like to tell that reviewer that survival analysis was carried out only for the identified 9 key genes (and not miRNAs). Therefore, as suggested, the corrections have been made in the revised text. The P value (0.05) has been provided in the text (Page 21).

“Survival analysis of obtained key genes was undertaken using GEPIA. The overall survival analysis of the obtained key genes (CPEB3, SAMD8, FOXP1, TRPS1, TCF4, TBL1XR1, SPRED1, CELF2 and CDK19) was examined to link their correlation with the prognosis of NSCLC (Figure 10). Survival curves are used to show the survival ability with time and survival rate (using P value 0.05). Moreover, GEPIA tool was utilized to validate the expression of key genes between control and lung cancer tissues (in LUSC cohort). It was determined that the miRNA expression of genes CDK19, SAMD8, TBL1XR1 and TRPS1 were significantly upregulated in LUSC dataset between lung cancer patients and controls (Figure 11). Moreover, the relation between key gene expression and pathological/tumor stage in NSCLC patients was estimated that revealed that the association of key genes with tumor stage NSCLC patients (Figure 12)”.

  1. The authors identified several key miRNA target genes to be significantly associated with NSCLC, however, some of them are common targets by up-regulated and down-regulated miRNAs, which makes the results are less persuasive. For example, CDK19 is targeted by miR-130b, miR-205, miR-30a, miR-140-3p, miR-144.

Response 6. The identified hub miRNAs and its associated genes (by comparison of three datasets) from the present findings show consistency with several published literatures. Thus, in this regard, it could be interpreted that the present study results show validation of our retrieved data.

  1. Many typos were found in the manuscripts.

Response 7. We thank the reviewer for highlighting this. The corrections have been made in the revised text.

With warm regards

Anwar Ahmed

Associate Professor

Centre of Excellence in Biotechnology Research

College of Science

King Saud University

Riyadh (KSA)

Email: anahmed@ksu.edu.sa

Reviewer 2 Report

In this research paper, Zoya Shafat et al. carried out a comprehensive bioinformatic study with the aim of identifying key miRNAs and their target genes in Non-Small Cell Lung Cancer. Overall I think the work deserves publication in the Genes journal, nonetheless some issues must be addressed. At first, the manuscript contains an enormous number of typing errors (see further comments, but recheck the whole manuscript as I certainly didn't identify them all). Secondly, many figures are blurred and some of them are impossible to read, so it is necessary to upload high-resolution figures to the final version of the paper. Further comments are described below:

Major comments:

I was unable to find any supplementary material (S1 file), but it is mentioned at the end of the manuscript, please check

The 7-page long Table 2 should be placed in the Supplementary materials, there is no justification for its appearance in the main text.

Figure 2 ... "diagram showing overlapping genes (4186) between" ... where does the number 4168 come from? Is it correct? Because it doesn't correspond with the Figure 2 content (?)

Figure 6 ... please explain your centrality measures in detail (Degree, Closeness, Betweenness, and Stress

Figure 13 ... what does "outlier" (red stars) mean here statistically? Please specify

Figure 14 ... y-axis must be described and explained

Minor points:

Line 25 ... gens should be genes

Line 68 ... gene expression omnibus ... GEO abbreviation should be used here first

Line 93 ... inclusion ofmicroRNA (miRNA) ... space missing

Line 103 ... generated fromGPL7731 ... space missing

Line 107 ... ung should be lung

Line 110 ... DEMS should be DEMs

Line 115 ... adjusted p-value ... please specify (Benjamini Hochberg, or Bonferroni, or something other?)

Line 133 ... CTD ... please explain the abbreviation

Line 142 ...wereanalyzed

Line 144 ... nodereflects

Line 152 ... allnodes

Line 153 ... Pathway Analysis (bigger text font, why?)

Line 155 ... add links where you are talking about web-tools

Line 168 ... redundant "O"

Line 175 ... ofDEGs

Line 190 - 194 ... why in bold?

Line 215 ... Overalpped genes (2x)

Line 230 ... obtained form 4 datasbases ... form should be from ... typo in datasbases

Figures 3,4,5 ... what is their significance? It should be discussed better in the text. Also, are there any p-values supporting these results? Maybe better to place Figures 3 and 4 into supplementary materials and in the main text present only Figure 5...

Line 261 ... 13common

Line 262 ... ninegenes

Please double-check gene and protein names in the whole manuscript ... genes should be always written in Italics (e.g. p63 is the protein and TP63 is the gene) ... this should be unified

Line 350 ... theTFs 

Line 352 ... enrichr database

Line 354 ... assciated TFs

Line 396 ... "regulated (miR 30a,miR 145, miR 140 3p, miR 572, miR 144, miR 126 and miR 486" ... should be "miR30a" or "miR-30a", etc. ... please check all miRNA names in the manuscript and unify

Line 404 and 405 ... why in bold and bigger font used?

Line 420 and 437 ...why in the bold and bigger font/Italics used?

Line 491 ... why in the bold and bigger font used?

Author Response

Author’s point by point response to the Reviewer comments

We have seen the comments and noted the concerns raised by the Editor and Reviewers. In this context we have provided some additional information in the point-by-point response to make it more explicit. We also appreciate the efforts of the Editor and esteemed reviewers whose comments have indeed uplifted the quality of our manuscript.

Reviewers Comments and Author’s Response:

Reviewer 2

Comments and Suggestions for Authors In this research paper, Zoya Shafat et al. carried out a comprehensive bioinformatics study with the aim of identifying key miRNAs and their target genes in Non-Small Cell Lung Cancer. Overall I think the work deserves publication in the Genes journal, nonetheless some issues must be addressed. At first, the manuscript contains an enormous number of typing errors (see further comments, but recheck the whole manuscript as I certainly didn't identify them all). Secondly, many figures are blurred and some of them are impossible to read, so it is necessary to upload high-resolution figures to the final version of the paper. Further comments are described below:

Response. We would like to thank the reviewer for appreciating our work and providing constructive feedback. At first, we would like to tell the reviewer that the typing errors in the overall text have been corrected. Secondly, as suggested, most of the figures with better resolution have been incorporated in the revised version of the manuscript.

Major comments:

  1. I was unable to find any supplementary material (S1 file), but it is mentioned at the end of the manuscript, please check.

Response 1. We thank the reviewer for pointing-out this mistake. The S1 table has been now incorporated within the revised text.

  1. The 7-page long Table 2 should be placed in the Supplementary materials, there is no justification for its appearance in the main text.

Response 2. We thank the reviewer highlighting this. As suggested, Table 2 has been removed from the main text and added as a supplementary file (Additional File: S2) within the revised text.

  1. Figure 2 ... "diagram showing overlapping genes (4186) between" ... where does the number 4168 come from? Is it correct? Because it doesn't correspond with the Figure 2 content (?)

Response 3. Thank you for highlighting this. Yes, the number 4186 is correct. The number 4186 is the total of the column OG vs CTD, i.e., the last column, in Table 2 (Page 6).

The paragraph has further been incorporated in the revised text (Page 7).

“We identified the target genes for the top 12 DEMs using a combination of four databases, i.e., mirMap, TargetScan, miRWalk and mirDIP. Each of these databases showed different target genes for each of the input miRNAs. We selected only those target genes which were given by at least two of these databases and excluded those target genes which were validated by only one of these databases. Based on this selection criterion, we obtained a total of 4186 target genes for the top 12 DEMs. Figure 2 is the Venn diagram representation of the results given by these 4 databases, for example, the value “152” shown in green represents the number of target genes given by both mirDIP and miRWalk (Figure 2).”

The legend of Figure 2 has been revised and now reads “Venn diagram showing overlapping genes between ...”

  1. 4. Figure 6 ... please explain your centrality measures in detail (Degree, Closeness, Betweenness, and Stress.

Response 4. The detail about centrality measures have been added in the revised text (Pages: 4 -5).

“Degree distribution: In a particular network, the degree (k) of node reflects the total number of edges (connections) by which it is connected with other nodes [26]. The degree k of a node is a local measure of centrality of that node [raman]. The degree distribution P(k) of a node n is given by the expression:

where, nk is the number of nodes having degree k and N is the total number of nodes in the network. P(k) indicates the importance of hubs or modules in the network.

Betweenness centrality: In a particular network, a node’s betweenness centrality reflects the importance of flow of information from one node to another based on the shortest path [27]. The betweenness centrality CB(n) of a node n is given by the expression:

where, s and t are nodes in the network other than ndst is the total number of shortest paths from s to t, and dst (n) is the number of those shortest paths from s to t on which n lies [26, 28, 29].

Closeness centrality: In a particular network, closeness centrality reflects how the information is rapidly passing from one node to another [30].

where, dij is the length of the shortest path between two nodes i and j, and N is the total number of nodes in the network which are connected to the node n.

Stress: In a particular network, stress reflects the addition of all nearest path of all node pairs [31]. In order to compute the stress of a node v, first calculate all shortest pathways in a graph G, then, count the number of shortest paths passing through v. A stressed node is the one that has large number of shortest paths passing through it. Notably, and may be more critically, a high stress number does not necessarily imply that node v is critical for maintaining the link between nodes whose pathways cross through it.

  1. Figure 13 ... what does "outlier" (red stars) mean here statistically? Please specify.

Response 5. As suggested, the meaning has been added in the legend of Figure 11 (Page 25).

“….Outlier’ is regarded as fringe elements having extra values from the whole data”.

  1. Figure 14 ... y-axis must be described and explained

 Response 6. As suggested, the meaning has been added in the legend of Figure 12 (Page 27).

     “The X-axis represents the major pathological stages while the Y-axis represents the log scale transformed expression data”.

Minor points:

Line 25 ... gens should be genes

Response 1. The sentence has been corrected and reads “The key genes and…..” (Page 1).

  1. Line 68 ... gene expression omnibus ... GEO abbreviation should be used here first

Response 2. As suggested, the sentence has been revised and reads “Three GEO (Gene Expression Omnibus) datasets…” (Page 2).

  1. Line 93 ... inclusion of microRNA (miRNA) ... space missing

Response 2. As suggested, the sentence has been revised and reads “The criteria for inclusion of miRNA…” (Page 3).

  1. Line 103 ... generated from GPL7731 ... space missing

Response 4. The needful was done as per the suggestion (Page 3).

  1. Line 107 ... ung should be lung

Response 5. As suggested, the sentence has been revised and reads “….dataset consisted of 55 lung cancer…..” (Page 3).

  1. Line 110 ... DEMS should be DEMs

Response 6. As suggested, the sub-heading 2.3 has been revised and reads “Data Preprocessing and Screening of Differentially Expressed miRNAs (DEMs)” (Page 4).

  1. Line 115 ... adjusted p-value ... please specify (Benjamini Hochberg, or Bonferroni, or something other?)

Response 7. The line has been added “Benjamini-Hochberg correction method was used to correct the significant P-values obtained by the original hypothesis test” (Page 4).

  1. Line 133 ... CTD ... please explain the abbreviation

Response 8. As suggested, the abbreviation of CTD has been incorporated in the revised version of the manuscript. Please see Page 4.

“The miRNA–mRNA network was built by utilizing overlapped genes [target genes vs. CTD (Comparative Toxicogenomics Database) NSCLC genes]…..”.

  1. Line 142 ...were analyzed

Response 9. As suggested, the sentence has been revised and reads “….network were analyzed to……” (Page 4).

  1. Line 144 ... node reflects

Response 10. The needful was done as per the suggestion (Page 4).

  1. Line 152 ... all nodes

Response 11. The sentence now reads as “…nearest path of all node pairs” (Page 5).

  1. Line 153 ... Pathway Analysis (bigger text font, why?)

Response 12. As suggested, the correction has been done in the revised text (Page 5).

  1. Line 155 ... add links where you are talking about web-tools

Response 13. The needful was done as per the suggestion (Page 5).

“using MIENTURNET (MIcroRNAENrichmentTURnedNETwork) web-tool (Mienturnet (uniroma1.it)) [32] …..”

  1. Line 168 ... redundant "O"

Response 14. The needful was done as per the suggestion (Page 5).

“The survival analysis was undertaken by constructing the overall survival (OS) curve of key genes. The patients …..”.

  1. Line 175 ... ofDEGs

Response 15. The sentence now reads as “Selection of DEGs” (Page 6).

  1. Line 190 - 194 ... why in bold?

Response 16. Thank you for highlighting this. The needful was done as per the suggestion (Page 6).

  1. Line 215 ... Overalpped genes (2x)

Response 17. Thank you for highlighting this. The needful was done as per the suggestion (Page 7).

“Log FC: Log fold cange; OG: Overlapped genes; CTD: Comparative Toxicogenomics Database. The overlapped genes were predicted by 4 databases: TargetScan, miRWalk, mirDIP, and miRmap.

”.

  1. Line 230 ... obtained form 4 datasbases ... form should be from ... typo in datasbases

Response 18. Thank you for highlighting this. The sentence now reads as “Target genes (obtained from 4 databases)” in the revised text (S2 File).

  1. Figures 3,4,5 ... what is their significance? It should be discussed better in the text. Also, are there any p-values supporting these results? Maybe better to place Figures 3 and 4 into supplementary materials and in the main text present only Figure 5...

Response 19. Thank you for highlighting this. As suggested, Figure 2 and Figure 3 in the older version of the manuscript have now been moved to supplementary files as Additional File: S3 and  Additional File: S4, respectively. No p-values supporting these results. Further, more detail about these figures have been incorporated in the results section 3.4 (Pages: 8 - 9).  

“The construction of miRNA-mRNA network using overlapped genes (target genes vs CTD (Comparative Toxicogenomics Database) NSCLC genes) build from SIF files. The up and down regulated network were separately built by Cytoscape as shown in the additional material (Additional File: S3) and (Additional File: S4) respectively. The upregulated miRNA-target gene interaction network contained 1728 nodes and 1928 edges, wherein, triangles (green) represented the upregulated miRNAs and circles (blue) represented the interacting gene partners. The downregulated miRNA-target gene interaction network contained 1895 nodes and 2256 edges, wherein, diamonds (red) represented the downregulated miRNAs and circles (blue) represented the interacting gene partners. In both upregulated and downregulated miRNA-mRNA network, the target genes were obtained from different databases and the common ones were preceeded forward”.

“The merge interaction network constructed using upregulated and downregulated DEMs is represented in Figure 3. The merged network was constructed using Cytoscape software. Further, this built merged network was used for analysis of modules detection. This is the new way to construct the miRNAs-mRNAs network by using SIF Files. If the network was constructed uisng web tools like miRNet, Network Analyst and MIENTURNET, than all key miRNAs would not have been interacted. In this regard, to find target genes for each miRNA we utilized four different databases (to validate our results, different databases were used to cross-check the target genes). Thus, after obtaining all the overlapped target genes (from 4 databases), these were further used for construction of the network”.

  1. Line 261 ... 13 common

Response 20. As suggested the sentence has been revised and reads, “The obtained 13 common…” (Page 11).

  1. Line 262 ... nine genes

Response 21. The needful was done as per the suggestion (Page 11).

“…nine genes….”

  1. Please double-check gene and protein names in the whole manuscript ... genes should be always written in Italics (e.g. p63 is the protein and TP63 is the gene) ... this should be unified

Response 22. Thank you for highlighting this. The needful was done as per the suggestion (Page 31). Other corrections have also been made in the revised text.

“Moreover, the expression plots of the identified key genes showed significant correlation with NSCLC prognosis. On combining these results, it could be interpreted that the key genes played significant role in the regulation of NSCLC. For further elucidation, the key genes were predicted for their TFs. TP63, a member of tumor suppressor protein p53 [105], is known to associate to development and tumorigenesis of cancers, in particular to cancer metastasis [106]. VHL is a product of the tumor suppressor gene VHL. Recently, a study on human kidney cells has shown its function in the cell growth regulation and differentiation [107]. LEF1, a protein encoded by LEF1 gene in humans, has shown its expression in several cancers [108]. This protein belongs to TCF (T-cell Factor) family, thus is involved in Wnt signaling pathway and is vital for stem cell maintenance and organ development [109]. RUNX3, a protein encoded by RUNX3 gene in humans and a component of TGF-β (transforming growth factor-β), has shown tumor suppressive effects in several cancers [110, 111]. ESR1, a protein encoded by ESRI gene, has shown its association to many kinds of cancers (endometrial, breast and prostate) [112]. EGR1 is chiefly involved in tissue injury, fibrosis and immune response processes. Recent reports have shown the involvement of EGR1 in the initiation and succession of cancer. Nevertheless, the precise mechanism of EGR1 modulation remains unelucidated [113]. AR (Androgen receptor), a ligand-dependent transcription factor, has been shown to involve in protrate cancer [114]. Thus, it could be interpreted that these identified TFs may play significant role in the pathogenesis of NSCLC”.

  1. Line 350 ... theTFs 

Response 23. The sentence now reads as “Identification of the TFs” (Page 27).

  1. Line 352 ... enrichr database

Response 24. Thank you for highlighting this. The needful was done as per the suggestion (Page27).

  1. Line 354 ... assciated TFs

Response 25. The sentence has been revised and reads “…associated TFs…” (Page 27).

  1. Line 396 ... "regulated (miR 30a,miR 145, miR 140 3p, miR 572, miR 144, miR 126 and miR 486" ... should be "miR30a" or "miR-30a", etc. ... please check all miRNA names in the manuscript and unify

Response 26. We thank the reviewer for highlighting this. As suggested, the corrections have been made throughout the revised manuscript.

  1. Line 404 and 405 ... why in bold and bigger font used?

Response 27. Thank you for highlighting this. The corrections have been made in the revised text (Page 29).

  1. Line 420 and 437 ...why in the bold and bigger font/Italics used?

The needful was done as per the suggestion in the discussion section (Page 29).

  1. Line 491 ... why in the bold and bigger font used?

Response 29. As suggested, the needful was done in the revised text (Page 31).

With warm regards

Anwar Ahmed

Associate Professor

Centre of Excellence in Biotechnology Research

College of Science

King Saud University

Riyadh (KSA)

Email: anahmed@ksu.edu.sa

Round 2

Reviewer 1 Report

The authors have addressed most of the questions that I have.

Author Response

Author’s point by point response to the Reviewer comments

We have seen the comments and noted the concerns raised by the Editor and Reviewers. In this context we have provided some additional information in the point-by-point response to make it more explicit. We also appreciate the efforts of the Editor and esteemed reviewers whose comments have indeed uplifted the quality of our manuscript.

Reviewers Comments and Author’s Response:

Reviewer 1

Comments and Suggestions for Authors The authors have addressed most of the questions that I have.

Response. We would like to thank the reviewer for providing constructive feedback.

With warm regards

Anwar Ahmed

Associate Professor

Centre of Excellence in Biotechnology Research

College of Science

King Saud University

Riyadh (KSA)

Email: anahmed@ksu.edu.sa

Reviewer 2 Report

The manuscript was significantly improved. I have only a few minor points:

1.) Figure 1 - MiRNA should be miRNA and not underlined by a red line (Word autocorrect)

2.) Line 230 ... literatures should be literature

3.) Table 2 ... P value should be P-value, remove yellow highlighting

4.) Figure 2 legend ... remove yellow highlighting

5.) Line 293 ... preceeded (?)

6.) Figure 13 ... I still believe that the red asterisk expresses the statistical difference between two boxplots here, please double-check

Author Response

Author’s point by point response to the Reviewer comments

We have seen the comments and noted the concerns raised by the Editor and Reviewers. In this context we have provided some additional information in the point-by-point response to make it more explicit. We also appreciate the efforts of the Editor and esteemed reviewers whose comments have indeed uplifted the quality of our manuscript.

Reviewers Comments and Author’s Response:

Reviewer 2

Comments and Suggestions for Authors The manuscript was significantly improved. I have only a few minor points:

Response. We would like to thank the reviewer for appreciating our work and providing constructive feedback.

  1. MiRNA should be miRNA and not underlined by a red line (Word autocorrect)

Response 1. Thank you for highlighting this. As suggested, the word MiRNA has been revised and now reads “miRNA” (Page 2).

 “The microRNAs (miRNAs) are small (~22 nucleotides) noncoding RNAs which regulate more than half of the genes in human cells [9]. An miRNA is linked with diverse biological activities which include cell differentiation, cell proliferation, disease initiation, cell migration, disease progression and finally apoptosis [9]. The miRNA modulates the activity of gene at the level…”.

  1. Line 230 ... literatures should be literature

Response 2. As suggested, the word “literatures” has been revised and now reads “literature” (Page 6).

  1. Table 2 ... P value should be P-value, remove yellow highlighting

Response 3. Thank you for highlighting this. The needful was done as per the suggestion (Page 6).

“…identified on the basis of fold change (> 1.5) and P-value (< 0.05)”.

Table 2. Top 12 DEMs (7 downregulated and 5 upregulated) on the basis of log fold change and P-value…”.

  1. 4. Figure 2 legend ... remove yellow highlighting

Response 4. We thank the reviewer for highlighting this. The changes have been made in the legend of Figure 2 in the revised text (Page 8).

  1. Line 293 ... preceeded (?)

Response 5. As suggested, the corrections have been made in the revised text (Page 8).

“...common ones were proceeded further….”

  1. Figure 13 ... I still believe that the red asterisk expresses the statistical difference between two boxplots here, please double-check

Response 6. We thank the reviewer for pointing-out this mistake. The legend of Figure 13 has been revised and reads “…Outlier’ represents the statistical difference of gene expression between two boxplots” (Page 25).

With warm regards

Anwar Ahmed

Associate Professor

Centre of Excellence in Biotechnology Research

College of Science

King Saud University

Riyadh (KSA)

Email: anahmed@ksu.edu.sa
